# Medical students' perceptions and motivations during the COVID-19 pandemic

Patricia Tempski[1], Fernanda M. Arantes-Costa[1], Renata Kobayasi[1], Marina A. M. Siqueira[1], Matheus B. Torsani[1], Bianca Q. R. C. Amaro[2], Maria Eduarda F. M. Nascimento[3], Saulo L. Siqueira[1], Itamar S. Santos[1], Milton A. Martins[1] *

1 Centro de Desenvolvimento de Educação Médica e Departamento de Clínica Médica, Faculdade de Medicina da Universidade de São Paulo, São Paulo, Brazil, 2 Centro de Desenvolvimento de Educação Médica e Departamento de Clínica Médica, Universidade Federal de Roraima, Boa Vista, Brazil, 3 Centro de Desenvolvimento de Educação Médica e Departamento de Clínica Médica, Universidade de Pernambuco, Recife, Brazil

* mmartins@usp.br

## Abstract

### Background

There has been a rapid increase in the number of cases of COVID-19 in Latin America, Africa, Asia and many countries that have an insufficient number of physicians and other health care personnel, and the need for the inclusion of medical students on health teams is a very important issue. It has been recommended that medical students work as volunteers, undergo appropriate training, not undertake any activity beyond their level of competence, and receive continuous supervision and adequate personal protective equipment. However, the motivation of medical students must be evaluated to make volunteering a more evidence-based initiative. The aim of our study was to evaluate the motivation of medical students to be part of health teams to aid in the COVID-19 pandemic.

### Methods and findings

We developed a questionnaire specifically to evaluate medical students' perceptions about participating in the care of patients with suspected infection with coronavirus during the COVID-19 pandemic. The questionnaire had two parts: a) one part with questions on individual characteristics, year in medical school and geographic location of the medical school and b) a second part with twenty-eight statements assessed on a 5-point Likert scale (totally agree, agree, neither agree nor disagree, disagree and totally disagree). To develop the questionnaire, we performed consensus meetings with a group of faculty and medical students. The questionnaire was sent to student organizations of 257 medical schools in Brazil and answered by 10,433 students. We used multinomial logistic regression models to analyze the data. Statements associated with greater odds ratios for participation of medical students in the COVID-19 pandemic were related to a sense of purpose or duty ("It is the duty of the medical student to put himself or herself at the service of the population in the pandemic"), altruism ("I am willing to take risks by participating in practice in the context of the pandemic"), and perception of good performance and professional identity ("I will be a

**Data Availability Statement:** All relevant data are within the paper and its Supporting information files.

**Funding:** The authors received no specific funding for this work.

**Competing interests:** No authors have competing interests.

better health professional for having experienced the pandemic"). Males were more prone than females to believe that only interns should participate in the care of patients with COVID-19 (odds ratio 1.36 [coefficient interval 95%:1.24–1.49]) and that all students should participate (OR 1.68 [CI:1.4–1.91]).

## Conclusions

Medical students are more motivated by a sense of purpose or duty, altruism, perception of good performance and values of professionalism than by their interest in learning. These results have implications for the development of volunteering programs and the design of health force policies in the present pandemic and in future health emergencies.

## Introduction

The COVID-19 pandemic is the most important global health crisis of our time and the greatest challenge the health system has faced since World War Two. Since its emergence in Asia in 2019, the virus has spread to every continent except Antarctica. Cases are increasing daily in Europe, North America, and, in the last weeks, also in Latin America and Africa [1, 2].

The COVID-19 pandemic has resulted in a disruption of undergraduate medical education. In many countries, medical education faculty have quickly transitioned the first-year curriculum to online activities in response to the need for social isolation to flatten the curve of new cases of COVID-19 [3]. In addition, in the final years of medical schools, in many countries, clerkships have been severely affected by the rapid changes in hospitals due to the need for care of an increasing number of COVID-19 patients, and medical students have been advised to stay at home given the potential risk of medical students spreading COVID-19 infection in health care settings and the shortage of personal protective equipment (PPE) [3, 4].

However, the role of medical students in the COVID-19 pandemic is changing rapidly due to the shortage of health professionals in many cities, even in developed countries. Both the Medical Schools Council (MSC) of the United Kingdom and the American Association of Medical Colleges (AAMC) of the United States have published guidelines for the participation of medical students in the global effort to provide the best care to patients with COVID-19 [5, 6]. Both associations recommend that medical students work as volunteers, undergo appropriate training, not undertake any activity beyond their level of competence, and receive continuous supervision and adequate PPE [5, 6].

In recent weeks, there has been a very large increase in the number of cases in countries that have an insufficient number of physicians and other health care personnel, and it is possible to anticipate the need for the inclusion of medical students as part of health teams [2, 7].

In many countries, thousands of medical students have volunteered their services to support the fight against the coronavirus pandemic [8, 9]. Motivation is pivotal for volunteering and must be evaluated to make volunteering a more evidence-based initiative.

The aim of our study was to evaluate the motivation of medical students to be part of health teams to help in the COVID-19 pandemic. The study was performed in a developing country, Brazil, in the first week of the increase in cases of COVID-19 in Brazil and included 10,433 medical students.

## Materials and methods

### Ethics

Our study was approved by the Ethics Committee of the School of Medicine of the University of São Paulo (Comissão de Ética para Análise de Projetos de Pesquisa—CAPPesq) and by the National Committee of Ethics in Research of the Ministry of Health of Brazil (Comissão Nacional de Ética em Pesquisa—CONEP), protocol 3.990.128 and was developed according to the STROBE guidelines for cross-sectional studies. On the first page of the questionnaire, the purpose of the study was explained, and participating students had to complete a consent form. We guaranteed both confidentiality and anonymity, and the students could contact the research group by email if they wanted.

### Development of questionnaire

We developed a questionnaire specifically to evaluate medical students' perceptions about participating in the care of patients with suspected infection with coronavirus during the COVID-19 pandemic. The questionnaire had two parts: a) one part with questions on individual characteristics, year in medical school and the geographic location of the medical school and b) a second part with twenty-eight statements rated on a 5-point Likert scale (totally agree, agree, neither agree nor disagree, disagree and totally disagree).

To develop the questionnaire, we held four meetings, including two meetings with medical students and two meetings with faculty from the Center for Development of Medical Education of the School of Medicine of the University of Sao Paulo. Each meeting had 10–15 participants and lasted 2–3 hours. We asked the following questions to the participants:

"To assess motivation of medical students to work with health teams in the care of people with COVID-19, which questions should be asked?"

"What are the main concerns of medical students related to working with people with COVID-19?"

"What questions should be asked to evaluate the opinions and motivation of medical students concerning the need to move to online teaching during the COVID-19 pandemic?"

We recorded the meetings, and three researchers wrote the questionnaire, including all suggestions from the medical students and faculty meetings. The questionnaire was then revised by a group of medical students and faculty members until a consensus was reached that all the statements were easy to understand. A second revision was performed by the same group after a pilot application of the questionnaire with a group of twenty medical students.

### Data collection

The questionnaire was developed on the survey administration app Google Forms. With the help of the Brazilian Section of the International Federation of Medical Students Associations (IFMSA), the questionnaire was sent to student organizations of medical schools of all regions of Brazil using Instagram and WhatsApp.

The survey was performed between 20th and 22nd March; we closed the survey on 22nd March because we had received more than 10,000 answers after these three days of survey.

In this period, there were only a small number of patients diagnosed with COVID-19 in Brazil, including 1,546 confirmed cases and 25 deaths due to COVID-19, according to the Ministry of Health of Brazil [10].

In Brazil, undergraduate medical programs are six years, including four years of basic and clinical sciences and two years of clerkships (internship), when medical students have

responsibilities in the direct care of patients under the supervision of faculty or preceptors of the National Health System.

There are 341 medical schools in Brazil with approximately 35,288 first-year medical students [11]. The questionnaire was sent to the student organizations of 257 medical schools (75.4% of Brazilian medical schools).

## Study variables

The participants indicated their degree of agreement with two statements regarding their views about the role that medical students should have during the COVID-19 pandemic. These statements included the following: (S8) Medical internship students must participate in health care assistance during the pandemic and (S9) All students, regardless of their year in medical school, must participate in health care assistance during the pandemic. Participants were classified according to their agreement with statements S8 and S9 as follows: (1) All students should participate (agree with S9); (2) Only students in internships should participate (neither agree nor disagree/disagree with S9 and agree with S8); and (3) No students should participate (neither agree nor disagree/disagree with S9 and with S8).

For the analysis, we combined the responses "completely agree" with "agree" for each statement.

## Statistical analysis

Continuous variables are expressed as the means ± SD. Categorical variables are expressed as absolute counts and proportions, and they were compared across groups using chi-squared tests.

For exploratory factor analysis (EFA) and regression models, agreement with each statement was considered as a dichotomous variable. In these cases, participants who answered "totally agree" or "agree" were considered as agreeing with each statement. We performed an EFA using varimax rotation to identify the underlying latent variables (factors) in our questionnaire. The criteria for EFA model selection were as follows: (1) models with sum of squared loadings greater than 1 for all latent variables were initially considered, and (2) among these models, we selected the model with the highest number of factors. This process led to the selection of a model with four factors. Items with loadings less than -0.3 or greater than 0.3 were considered relevant for each factor. To compute each EFA latent variable score, we attributed weights for all items, according to their loadings in the final model, regardless of whether they met the criteria for relevance. After this procedure, we standardized the values to obtain a mean of zero and standard deviation (SD) of one for each factor score. Therefore, the odds ratios presented in these models relate to a 1 SD increase in the EFA latent variable score. In this model, an odds ratio of 2 represents twice the chance of a respondent have a score one standard deviation higher than the mean score of the factor.

We used multinomial logistic regression models to study the association between the students' characteristics and perceptions according to their opinion about student participation in the healthcare of the COVID-19 pandemic. The medical students were divided into three groups according to their opinions on the participation of medical students in the care of people with COVID-19: a) no students should participate (reference); b) only interns (students in the last two years of medical school) should participate; and c) all medical students should participate. The presented models are (1) a crude model and (2) a model adjusted for sex, year in medical school and geographic region of the country where the medical school is located.

Analyses were performed using IBM Corp. Released 2013, SPSS Statistics for Windows, Version 22.0. Armonk, NY and R software version 3.4.4 (Vienna, Austria).

**Table 1. Participant distribution according to geographic region of Brazil.**

|  | Southeast | Northeast | South | Middle-West | North |
|---|---|---|---|---|---|
| **Participants (% of total participants)** | 5104 (48.9%) | 2456 (23.5%) | 1484 (14.2%) | 750 (7.2%) | 639 (6.1%) |
| **Vacancies (first year)** | 16190 | 8602 | 5329 | 3013 | 2702 |
| **Participants/vacancies** | 0.32 | 0.29 | 0.28 | 0.25 | 0.24 |

## Results

Table 1 shows the number of medical students from the five geographic regions of the country that answered the questionnaire and the relationship between the number of participants and the number of positions for first-year medical students (vacancies) in each region of the country. A similar proportion of respondents/vacancies was noted, demonstrating that the sample was homogenously distributed across the country.

Table 2 shows the number (and percentage) of medical students who answered "totally agree" or "agree" for each of the 28 statements of the questionnaire. Students were divided by year in medical school and sex. S1 Table shows the answers to the questionnaire for the entire group of medical students.

Of the 10,433 students who completed the survey, 7,267 (69.7%) were females, and 2,445 (23.4%) were in their internship years (interns). The mean age was 22.5 ± 3.9 years.

A total of 1,398 (13.4%) participants believed that all students should participate in the response to the COVID-19 pandemic, 4,963 (47.6%) participants believed that only students in internships should participate, and 4,072 (39.0%) participants believed no students should participate.

S2 Table presents the participants' characteristics and perceptions according to their views on the role of medical students during the COVID-19 pandemic.

Factor analysis of the questionnaire resulted in the identification of four factors (domains) that we referred to as remote learning, medical knowledge self-efficacy, psychological stress and professional values/altruism (S1 Table).

Table 3 shows the adjusted odds ratios (and 95% confidence intervals) from multinomial models for the association between students' characteristics and perceptions and their views on the role of medical students during the COVID-19 pandemic (the crude model results are presented in S4 Table). Compared to those for the participation of no medical students, the odds ratios [ORs] for the participation of students who are in internships or all medical students were 0.61 (95% confidence interval: 0.55–0.67) and 0.13 (95% CI: 0.10–0.16), respectively. Men were more prone to support the participation of medical students in the fight against the pandemic (the ORs for the participation of students in internships and all medical students were 1.36 [95% CI: 1.24–1.49] and 1.68 [95% CI: 1.47–1.91], respectively).

We also included the multinomial models of the four domains of the questionnaire observed with factor analysis (Table 3). There was a strong positive association between domain four (professional values/altruism) and the support of the participation of medical students in internships or all medical students (ORs of 3.81 [3.59–4.06] and 12.64 [11.37–14.06] in adjusted models, respectively). We also observed a positive association of domain 2 (medical knowledge self-efficacy) and a negative association of domain 3 (psychological stress) with support of the participation of medical students.

The following beliefs regarding the participation of medical students in COVID-19 pandemic healthcare had the highest odds ratios:

**Table 2. Students' perceptions according to year in medical school and sex during the COVID-19 pandemic—The number of students who responded "totally agree" or "agree" (% of total).**

| | Year of graduation | | | Gender | |
|---|---|---|---|---|---|
| | 1st/2nd | 3rd/4th | 5th/6th | Female | Male |
| S1 I feel prepared to identify a patient with suspected infection*# | 2252 (49.3) | 2262 (66.1) | 1879 (76.9) | 4277 (58.9) | 2116 (66.8) |
| S2 I can identify signs of severity in a patient*# | 2721 (59.6) | 2760 (80.7) | 2169 (88.7) | 5182 (71.3) | 2468 (78.0) |
| S3 I know how to guide patients in preventive measures# | 4316 (94.5) | 3310 (96.8) | 2341 (95.7) | 6962 (94.9) | 3005 (95.8) |
| S4 I know how to guide patients in therapeutic measures*# | 984 (21.5) | 1037 (30.3) | 1239 (50.7) | 2156 (29.7) | 1104 (34.9) |
| S5 I know how to use personal protection equipment (PFE)# | 3615 (79.2) | 2773 (81.1) | 1867 (76.4) | 5749 (79.1) | 2506 (79.2) |
| S6 I am able to participate in the care of patients who seek health care*# | 849 (18.6) | 1230 (36.0) | 1370 (56.0) | 2206 (30.4) | 1243 (39.3) |
| S7 I feel able to communicate a diagnosis of COVID-19 infection*# | 1153 (25.2) | 1435 (41.9) | 1423 (58.2) | 2489 (34.3) | 1522 (48.1) |
| S8 Medical internship students must participate in health care assistance during pandemic*# | 3054 (66.9) | 2011 (58.8) | 1225 (50.1) | 4206 (57.9) | 2084 (65.8) |
| S9 All students, regardless of their year in medical school, must participate in health care assistance during pandemic*# | 830 (18.2) | 478 (14.0) | 90 (3.7) | 888 (12.2) | 510 (16.1) |
| S10 It is the duty of the medical student to put himself or herself at the service of the population in the pandemic*# | 2190 (48.0) | 1433 (41.9) | 853 (34.9) | 3005 (41.4) | 1471 (46.5) |
| S11 I feel insecure regarding the future*# | 2786 (61.0) | 2217 (64.8) | 1766 (72.2) | 5068 (69.7) | 1701 (53.7) |
| S12 I am afraid of contaminating myself*# | 2705 (59.2) | 2055 (60.1) | 1673 (68.4) | 4695 (64.6) | 1738 (54.9) |
| S13 Medical schools must suspend their academic activities during the first to fourth years# | 3365 (73.7) | 2563 (74.9) | 2197 (89.9) | 5670 (78.0) | 2455 (77.5) |
| S14 Medical schools must suspend their academic activities during internships*# | 1175 (25.7) | 1130 (33.0) | 1061 (43.4) | 2435 (33.5) | 931 (29.4) |
| S15 Distance learning must be implemented during the suspension of academic activities*# | 2322 (50.8) | 2186 (63.9) | 1634 (66.8) | 4443 (61.1) | 1699 (53.7) |
| S16 I would prefer to delay my training to fully replace academic activities than to participate in distance learning activities*# | 1813 (39.7) | 991 (29.0) | 715 (29.2) | 2304 (31.7) | 1215 (38.4) |
| S17 After the pandemic, academic activities must be fully resumed*# | 3017 (66.1) | 1917 (56.0) | 1178 (48.2) | 4125 (56.8) | 1987 (62.8) |
| S18 After the pandemic, only practical academic activities must be resumed*# | 1699 (37.2) | 1665 (48.7) | 985 (40.3) | 3167 (43.6) | 1182 (37.3) |
| S19 I feel able to study my medical course content through distance learning# | 1913 (41.9) | 1927 (56.3) | 1438 (58.8) | 3655 (50.3) | 1623 (51.3) |
| S20 I prefer to study theoretical content using distance learning methods*# | 1604 (35.1) | 1631 (47.7) | 1322 (54.1) | 3267 (45.0) | 1290 (40.7) |
| S21 My emotional state during the pandemic affects my learning*# | 1856 (40.6) | 1319 (38.6) | 1089 (44.5) | 3217 (44.3) | 1047 (33.1) |
| S22 I will be a better health professional for having experienced the pandemic*# | 2606 (57.0) | 1936 (56.6) | 1393 (57.0) | 4020 (55.3) | 1914 (60.5) |
| S23 I feel stressed in the hospital at the moment*# | 1168 (25.6) | 1215 (35.5) | 1226 (50.1) | 2675 (36.8) | 934 (29.5) |
| S24 The supervision I receive in my practice fields is good*# | 1874 (41.0) | 1820 (53.2) | 1187 (48.5) | 3324 (45.7) | 1557 (49.2) |
| S25 I have access to psychological support*# | 2201 (48.2) | 1431 (41.8) | 747 (30.6) | 2938 (40.4) | 1441 (45.5) |
| S26 I am proud of the way my institution responded to social and health demands in the face of the pandemic*# | 2653 (58.1) | 1905 (55.7) | 991 (40.5) | 3798 (52.3) | 1751 (55.3) |
| S27 The role of medical students during the pandemic is irrelevant*# | 185 (4.1) | 204 (6.0) | 275 (11.2) | 463 (6.4) | 201 (6.3) |
| S28 I am willing to take risks by participating in practice in the context of the pandemic*# | 1967 (43.1) | 1581 (46.2) | 1282 (52.4) | 3144 (43.3) | 1686 (53.3) |

Chi-square test, gender

*P<0.05

Chi-square test, year of graduation

#P<0.05

S10 "It is the duty of the medical student to put himself or herself at the service of the population in the pandemic",

S28 "I am willing to take risks by participating in practice in the context of the pandemic",

S6 "I am able to participate in the care of patients who seek health care",

S7 "I feel able to communicate a diagnosis of COVID-19 infection", and

S22 "I will be a better health professional for having experienced the pandemic".

**Table 3. Adjusted odds ratios (95% confidence intervals) for the association between students' characteristics and perceptions and their views on the role of medical students during the COVID-19 pandemic.**

| | No students should participate | Only students in internships should participate | All students should participate |
|---|---|---|---|
| **Year in medical school** | | | |
| First/second (Basic sciences) | 1.0 (Reference) | Reference | Reference |
| Third/fourth (Clinical sciences) | 1.0 (Reference) | **0.72 (0.65–0.79)** | **0.60 (0.53–0.69)** |
| Fifth/sixth (Internship) | 1.0 (Reference) | **0.61 (0.55–0.67)** | **0.13 (0.10–0.16)** |
| **Sex** | | | |
| Female | 1.0 (Reference) | 1.0 (Reference) | 1.0 (Reference) |
| Male | 1.0 (Reference) | **1.36 (1.24–1.49)** | **1.68 (1.47–1.91)** |
| **Personal/family/friend diagnosis of COVID-19** | | | |
| No | 1.0 (Reference) | 1.0 (Reference) | 1.0 (Reference) |
| Yes | 1.0 (Reference) | 0.91 (0.77–1.07) | 0.87 (0.68–1.12) |
| **Factor analysis latent variables** | | | |
| Factor 1—Transition to remote learning | 1.0 (Reference) | 1.02 (0.98–1.07) | 1.00 (0.94–1.07) |
| Factor 2—Medical self-efficacy | 1.0 (Reference) | **2.23 (2.11–2.35)** | **4.80 (4.41–5.21)** |
| Factor 3—Psychological stress | 1.0 (Reference) | **0.48 (0.45–0.50)** | **0.33 (0.30–0.35)** |
| Factor 4—Altruism | 1.0 (Reference) | **3.81 (3.59–4.06)** | **12.64 (11.37–14.06)** |
| **Beliefs in support of the participation of medical students in COVID-19 pandemic healthcare** | | | |
| S10. It is the duty of the medical student to put himself or herself at the service of the population in the pandemic | 1.0 (Reference) | **5.03 (4.55–5.56)** | **44.10 (36.25–53.65)** |
| S28. I am willing to take risks by participating in practice in the context of the pandemic | 1.0 (Reference) | **5.04 (4.57–5.55)** | **20.34 (17.32–23.90)** |
| S6. I am able to participate in the care of patients who seek health care | 1.0 (Reference) | **3.55 (3.18–3.96)** | **10.74 (9.23–12.50)** |
| S7. I feel able to communicate a diagnosis of COVID-19 infection | 1.0 (Reference) | **2.15 (1.96–2.37)** | **5.33 (4.63–6.12)** |
| S22. I will be a better health professional for having experienced the pandemic | 1.0 (Reference) | **2.13 (1.95–2.32)** | **3.56 (3.11–4.08)** |
| S4. I know how to guide patients in therapeutic measures | 1.0 (Reference) | **1.83 (1.66–2.02)** | **3.20 (2.79–3.67)** |
| S24. The supervision I receive in my practice fields is good | 1.0 (Reference) | **1.74 (1.60–1.89)** | **2.84 (2.50–3.23)** |
| S3. I know how to guide patients in preventive measures | 1.0 (Reference) | **2.39 (1.95–2.94)** | **2.33 (1.69–3.22)** |
| S1. I feel prepared to identify a patient with suspected infection | 1.0 (Reference) | **1.59 (1.45–1.74)** | **2.32 (2.02–2.65)** |
| S2. I can identify signs of severity in a patient | 1.0 (Reference) | **1.48 (1.34–1.64)** | **2.32 (1.99–2.70)** |
| S5. I know how to use personal protection equipment (PFE) | 1.0 (Reference) | **1.68 (1.52–1.86)** | **2.22 (1.87–2.62)** |
| S25. I have access to psychological support | 1.0 (Reference) | **1.47 (1.34–1.60)** | **2.12 (1.87–2.40)** |
| S17. After the pandemic, academic activities must be fully resumed | 1.0 (Reference) | 1.09 (1.00–1.18) | **1.39 (1.22–1.59)** |
| S26. I am proud of the way my institution responded to social and health demands in the face of the pandemic | 1.0 (Reference) | **1.29 (1.18–1.40)** | **1.36 (1.20–1.54)** |
| S18. After the pandemic, only practical academic activities must be resumed | 1.0 (Reference) | 1.07 (0.99–1.17) | **1.21 (1.07–1.38)** |
| S19. I feel able to study my medical course content through distance learning | 1.0 (Reference) | **1.10 (1.01–1.20)** | **1.19 (1.05–1.35)** |
| S16. I would prefer to delay my training to fully replace academic activities than to participate in distance learning activities | 1.0 (Reference) | **0.89 (0.82–0.98)** | 1.10 (0.97–1.26) |
| **Beliefs not related to/poorly related to the participation of medical students in COVID-19 pandemic healthcare** | | | |
| S20. I prefer to study theoretical content using distance learning methods | 1.0 (Reference) | 1.05 (0.97–1.15) | 1.04 (0.91–1.18) |
| S15. Distance learning must be implemented during the suspension of academic activities | 1.0 (Reference) | 0.99 (0.91–1.08) | 1.05 (0.93–1.20) |
| **Beliefs against the participation of medical students in COVID-19 pandemic healthcare** | | | |
| S14. Medical schools must suspend their academic activities during internships | 1.0 (Reference) | **0.17 (0.15–0.19)** | **0.26 (0.22–0.30)** |
| S13. Medical schools must suspend their academic activities during the first to fourth years. | 1.0 (Reference) | **0.73 (0.65–0.82)** | **0.26 (0.23–0.30)** |
| S23. I feel stressed in the hospital at the moment | 1.0 (Reference) | **0.43 (0.40–0.47)** | **0.31 (0.27–0.36)** |

*(Continued)*

**Table 3.** (Continued)

| | No students should participate | Only students in internships should participate | All students should participate |
|---|---|---|---|
| **S12. I am afraid of contaminating myself** | 1.0 (Reference) | **0.53 (0.49–0.58)** | **0.39 (0.34–0.44)** |
| **S27. The role of medical students during the pandemic is irrelevant** | 1.0 (Reference) | **0.26 (0.21–0.32)** | **0.60 (0.47–0.78)** |
| **S21. My emotional state during the pandemic affects my learning** | 1.0 (Reference) | **0.66 (0.61–0.72)** | **0.65 (0.57–0.74)** |
| **S11. I feel insecure regarding the future** | 1.0 (Reference) | **0.79 (0.72–0.86)** | **0.73 (0.64–0.83)** |

On the other hand, beliefs against the participation of medical students in COVID-19 pandemic healthcare with the lowest odds ratios included statements S13/S14 (regarding the suspension of academic activities), S11 "I feel insecure regarding the future", S21 "My emotional state during the pandemic affects my learning", and S27 "The role of medical students during the pandemic is irrelevant".

Analysis with only the students in their last years (interns) was also performed (Table 4). In this subgroup, male sex was also associated with the support of the participation of medical students. Similar to the findings of the entire sample, statements S10, S28, S6, S2, S4 and S22 represented beliefs in support of the participation of medical students with the highest odds ratios in this subgroup. On the other hand, agreement with the statements S13/S14 (regarding the suspension of academic activities), S23 "I feel stressed in the hospital at the moment", S12 "I am afraid of contaminating myself" and S27 "The role of medical students during the pandemic is irrelevant" were more relevant beliefs associated with disagreement with the participation of medical students in the actions during the pandemic. Crude models for this subgroup are presented in S5 Table.

## Discussion

In this study, we aimed to evaluate the motivations of medical students to be part of health teams in the care of patients in the context of the COVID-19 pandemic. We also aimed to understand students' desires related to continuing medical training or having online education during the pandemic. To our knowledge, no previous study has evaluated a large sample of medical students for these purposes. We believe that this study can contribute to the planning of medical education and work force organization during the COVID-19 pandemic as well as other future health emergencies. Although our sample was a convenience sample, we evaluated a large group of medical students (10,433), and a uniform distribution was noted among subjects across the country (Table 1).

We observed that sense of purpose or duty (moral values linked to medical profession) was the most important factor that influenced the desire to work during the pandemic followed by the willingness to take risk (altruism), the perception of good performance (medical knowledge self-efficacy) and a perception of building of professional identity. In other words, among the students who believed that they should work during the pandemic, the desire to help was stronger than their interest in learning during this health emergency.

When we performed this study, the number of COVID-19 cases in Brazil was low, and only 7.0% of participants (Table 3) had a family member or a friend with a diagnosis of COVID-19 or had a personal diagnosis of this disease [10]. We were able to study the factors that influence the desire to participate in the care of COVID-19 without a strong emotional influence of sick relatives or friends.

**Table 4. Adjusted odds ratios (95% confidence intervals) for the association between internship students' characteristics and perceptions and their views on the role of medical students during the COVID-19 pandemic.**

| | No students should participate | Only students in internships should participate | All students should participate |
|---|---|---|---|
| **Sex** | | | |
| **Female** | 1.0 (Reference) | 1.0 (Reference) | 1.0 (Reference) |
| **Male** | 1.0 (Reference) | **1.29 (1.08–1.54)** | **1.75 (1.13–2.73)** |
| **Personal/family/friend diagnosis of COVID-19** | | | |
| **No** | 1.0 (Reference) | 1.0 (Reference) | 1.0 (Reference) |
| **Yes** | 1.0 (Reference) | 0.79 (0.58–1.06) | 0.46 (0.17–1.29) |
| **Beliefs in support of the participation of medical students in COVID-19 pandemic healthcare** | | | |
| **S10. It is the duty of the medical student to put himself or herself at the service of the population in the pandemic** | 1.0 (Reference) | **7.18 (5.86–8.79)** | **22.93 (13.45–39.10)** |
| **S28. I am willing to take risks by participating in practice in the context of the pandemic** | 1.0 (Reference) | **12.24 (10.04–14.92)** | **13.26 (7.66–22.97)** |
| **S6. I am able to participate in the care of patients who seek health care** | 1.0 (Reference) | **6.86 (5.69–8.26)** | **6.29 (3.79–10.44)** |
| **S2. I can identify signs of severity in a patient** | 1.0 (Reference) | **3.31 (2.47–4.44)** | **5.70 (1.78–18.25)** |
| **S4. I know how to guide patients in therapeutic measures** | 1.0 (Reference) | **2.70 (2.28–3.20)** | **4.63 (2.84–7.57)** |
| **S22. I will be a better health professional for having experienced the pandemic** | 1.0 (Reference) | **3.76 (3.16–4.48)** | **4.46 (2.72–7.32)** |
| **S24. The supervision I receive in my practice fields is good** | 1.0 (Reference) | **2.75 (2.32–3.26)** | **4.25 (2.65–6.80)** |
| **S7. I feel able to communicate a diagnosis of COVID-19 infection** | 1.0 (Reference) | **3.30 (2.77–3.93)** | **4.06 (2.45–6.72)** |
| **S25. I have access to psychological support** | 1.0 (Reference) | **2.12 (1.77–2.55)** | **3.50 (2.26–5.43)** |
| **S5. I know how to use personal protection equipment (PFE)** | 1.0 (Reference) | **2.58 (2.10–3.15)** | **2.79 (1.53–5.10)** |
| **S3. I know how to guide patients in preventive measures** | 1.0 (Reference) | **5.40 (3.14–9.30)** | 2.27 (0.70–7.35) |
| **S26. I am proud of the way my institution responded to social and health demands in the face of the pandemic** | 1.0 (Reference) | 1.14 (0.97–1.35) | **1.73 (1.12–2.66)** |
| **S19. I feel able to study my medical course content through distance learning** | 1.0 (Reference) | 1.01 (0.86–1.20) | **1.68 (1.05–2.67)** |
| **S17. After the pandemic, academic activities must be fully resumed** | 1.0 (Reference) | 0.87 (0.73–1.02) | **1.64 (1.05–2.56)** |
| **S1. I feel prepared to identify a patient with suspected infection** | 1.0 (Reference) | **2.67 (2.17–3.28)** | 1.63 (0.96–2.75) |
| **Beliefs not related to/poorly related to the participation of medical students in COVID-19 pandemic healthcare** | | | |
| **S20. I prefer to study theoretical content using distance learning methods** | 1.0 (Reference) | 0.85 (0.72–1.00) | 1.01 (0.65–1.55) |
| **S18. After the pandemic, only practical academic activities must be resumed** | 1.0 (Reference) | 1.01 (0.86–1.20) | 1.40 (0.91–2.15) |
| **Beliefs against the participation of medical students in COVID-19 pandemic healthcare** | | | |
| **S13. Medical schools must suspend their academic activities during the first to fourth years.** | 1.0 (Reference) | 0.79 (0.59–1.04) | **0.21 (0.13–0.34)** |
| **S14. Medical schools must suspend their academic activities during internships** | 1.0 (Reference) | **0.16 (0.14–0.20)** | **0.29 (0.19–0.46)** |
| **S23. I feel stressed in the hospital at the moment** | 1.0 (Reference) | **0.28 (0.23–0.33)** | **0.44 (0.28–0.68)** |
| **S12. I am afraid of contaminating myself** | 1.0 (Reference) | **0.37 (0.31–0.44)** | **0.53 (0.33–0.84)** |
| **S27. The role of medical students during the pandemic is irrelevant** | 1.0 (Reference) | **0.14 (0.10–0.20)** | 0.65 (0.35–1.22) |
| **S21. My emotional state during the pandemic affects my learning** | 1.0 (Reference) | **0.49 (0.41–0.58)** | 0.74 (0.47–1.14) |
| **S16. I would prefer to delay my training to fully replace academic activities than to participate in distance learning activities** | 1.0 (Reference) | **0.69 (0.58–0.83)** | 1.00 (0.63–1.58) |
| **S15. Distance learning must be implemented during the suspension of academic activities** | 1.0 (Reference) | **0.81 (0.68–0.96)** | 1.03 (0.65–1.65) |
| **S11. I feel insecure regarding the future** | 1.0 (Reference) | **0.69 (0.58–0.83)** | 1.06 (0.64–1.76) |

When we compared the answers to the statements of the questionnaire provided by students in the first two years of medical course to those of students in the last two years (interns) (Table 2, univariate analysis), we observed many differences.

The largest differences in the percentages of agreement with the statements were related to the feeling of competency to take care of patients (differences of 26.7% to 37.7% for statements S1, S2, S4, S6 and S7). As expected, interns felt more confident concerning the identification of a patient with suspected infection, the identification of signs of severity, guidance of patients in therapeutic measures, participation in the care of patients and the communication of a diagnosis. In addition, interns were more prone to accept online learning and the interruption of both classes and internship activities (differences of 16.0% to 19.0%). In contrast, students from the first years were more insecure about the substitution of classes and practical activities by online learning (statements S16 and S17, differences of 10.5% and 17.9%, respectively). More students preferred to delay their training and fully resume their academic activities.

Although interns felt more secure concerning their competency to care for COVID-19 patients, many interns probably needed psychological support. In fact, more interns agreed with statement S23 ("I feel stressed in the hospital at the moment", difference of 24.5% compared to first-year medical students), and fewer interns agreed with statement S25 ("I have access to psychological support", difference of 17.6%). In addition, many students were afraid of contaminating themselves (statement S12, percentages of agreement of 59.2 and 68.4%) and considered that their emotional state affected their learning (statement S21, percentages of agreement of 40.6 and 44.5%).

To perform multivariate analysis, we decided to divide the medical students into three groups based on their responses regarding the participation of medical students in the COVID-19 pandemic (no participation, participation only by interns or participation by all students; Table 3) to better understand the factors that were most important in the student's decision to participate. Given that we observed many differences between the answers of students in the first years of the medical program and interns, we performed two analyses, one including all medical students and the second including only interns (Tables 3 and 4). Interestingly, we did not observe important differences when we compared the two analyses, suggesting that individual factors are more important than professional identity developed during the medical course. Perhaps the decision to be a volunteer in a health emergency such as the COVID-19 pandemic is more linked to an emotional or attitudinal decision than beliefs of self-efficacy or performance.

The statements that had greater odds ratios when the groups who thought "all students should participate" and "only students in internship should participate" were compared to group of students who thought "no students should participate" (reference, odds 1.0) were statements S10, S28, S6, S7 and S22 for the comparison with the "all students" group and statements S10, S28, S6, S2, S4 and S22 for the comparison with the "interns only" group.

We observed that sense of purpose or duty (moral values linked to medical profession) (S10 "It is the duty of the medical student to put himself or herself at the service of the population in the pandemic") was the most important factor that influenced the desire to work during the pandemic followed by the willingness to take risk (altruism) (S28 "I am willing to take risks by participating in practice in the context of the pandemic") and the perception of good performance (medical knowledge self-efficacy) (S6 "I am able to participate in the care of patients who seek health care" and S7 "I feel able to communicate a diagnosis of COVID-19 infection" for all medical students and S6, S2 "I can identify signs of severity in a patient" and S4 "I know how to guide patients in therapeutic measures" for interns). In addition to statements that suggested moral values, altruism and confidence in professional competence, in both analyses, statement S22 ("I will be a better health professional for having experienced the pandemic"),

which was related to the building of professional identity, had odds ratios greater than 2.0 in all comparisons. In other words, among the students who believed that they should work during the pandemic, the desire to help was stronger than their interest in learning during this health emergency. Allowing students to participate can reinforce important values, such as altruism, service in times of crisis, and solidarity with the profession, contributing to the building of professional identity [12].

Why people act or decide to serve others is important to better organize volunteerism in a pandemic. Clary et al. studied volunteers of different areas and observed six different motivation functions to be a volunteer: value (opportunities to express altruism and humanitarian values), understanding (opportunities to learn something new or to develop skills), social (opportunities to establish relationships), career (opportunities to career benefits), protective (scape from their negative feelings) and enhancement (opportunities to add self-esteem) [13]. Studying 2,017 volunteers, Guntert et al. divided these functions into two categories: self-determined and controlled motivation. They included the functions of altruism and humanitarian values and understanding motives in the self-determined category and other functions, such as enhancement, protective, social and career, in the controlled motivation category [14]. They concluded that self-determined motivation resulted in more volunteer satisfaction [14].

In our study, we observed that students with a higher sense of duty (S10) and altruism (S28) were more prone to engage in health care activities during COVID-19, which could be interpreted as self-determined motivation or values as a function of motivation. We also observed that the self-perception of competence (S6 and S7) was the third factor influencing motivation, and the fourth was the desire to learn with the experience of working in the pandemic. The desire to learn demonstrated by students can be interpreted as an understanding function following Clary et al. and as a self-determined motivation following Guntert et al. [13, 14]. Controlled motivation was not important in our data.

We also evaluated factors that can influence the decision to not participate in the COVID-19 health effort. We observed that predictors of not being a volunteer were beliefs that all educational activities should be suspended (S13 and S14), fear of contamination (S12 "I am afraid of contaminating myself") and emotional factors (S23 "I feel stressed in the hospital at the moment").

The differences we observed between male and female medical students were smaller than the differences among students of different years of medical programs (Table 2, univariate analysis). Differences between males and females concerning the percentage of agreement with the statements of the questionnaire were greater than 9% in only 5 statements (females vs males): S11 "I feel insecure regarding the future" (16.0%), S7 "I feel able to communicate a diagnosis of COVID-19 infection" (-13.8%), S21 "My emotional state during the pandemic affects my learning" (11.2%), S28 "I am willing to take risks by participating in practical activities in the context of pandemic" (-10.0%) and S12 "I am afraid of contaminating myself" (9.7%). All these statements are related to emotional competencies. We also performed multinomial regressions to compare male to female medical students concerning the agreement with the statements that "no students", "only interns" and "all students" should participate in COVID-19 pandemic (Tables 3 and 4). Males were more prone to believe that only interns should participate (odds ratios 1.36 and 1.29, respectively, for all students and interns) and that all students should participate (odds ratios 1.68 and 1.75, respectively). We argue that this difference is possibly due to males' higher propensity to take risks in the health/safety domain [15]. Additionally, females are more prone to develop anxiety and stress disorders; be more affected by human suffering; and have worse perceptions about their own quality of life, health and skills [16–18]. These factors can influence their intentions to act during a pandemic.

Nonetheless, we cannot disregard gender bias as a possible limitation of our instrument [18, 19].

Our study has some limitations. Our sample was not randomized, but it had more than 10,000 medical students from all regions of Brazil. The survey was performed at the beginning of the COVID-19 pandemic in Brazil, and the results could be different if similar surveys were performed in different phases of the pandemic.

The COVID-19 pandemic has had a substantial impact on medical education across the world [3, 5]. There is uncertainty and disagreement about the appropriate roles for medical students during this pandemic, and student participation in clinical care has varied across institutions and countries [4, 12]. Many medical schools have continued to forbid any patient interaction, others have included medical students in patient care, and others have decided to graduate medical students early to make them frontline clinicians. The American Association of Medical Colleges recommended that "unless there is a critical health care workforce need locally, we strongly suggest that medical students not be involved in any direct patient care activities" [4]. However, some medical educators have a different point of view, considering that medical schools should offer students clinical opportunities that would benefit patient care and potentially help to prevent workforce shortages [20, 21]. These different attitudes may have implications for medical education concerning the role of medical students in a local or global health emergency. Some educators chose the position to ensure student security and to avoid exposition, and this perspective can send a message (a hidden curriculum) that students have a passive role without significant social responsibility. On the other hand, medical educators assert that the inclusion of medical students in health teams sends a message to medical students that social responsibility is pivotal to professional identity. Interestingly, in our study, 56.9% of medical students agreed with statement S22 ("I will be a better health professional for having experienced the pandemic").

A shortage of health professionals has occurred, even in some cities in developed countries [20, 21]. This risk will become even greater as the pandemic reaches developing countries in South America, Asia and Africa. The participation of medical students in the care of people with suspected or confirmed COVID-19 increases their personal risk of acquiring this disease. However, their risks of severe disease are probably lower than those of retired clinician volunteers, who are more susceptible to complications of COVID-19 given their age [22]. As personal risks cannot be eliminated, there is predominant agreement that the involvement of medical students in the care of patients should be voluntary [5, 6]. Medical students who work as volunteers must have appropriate training, must not undertake any activity beyond their level of competence, and must receive continuous supervision and adequate personal protective equipment [5, 6].

One implication of our study for medical education is that allowing students to participate in pandemic efforts reinforces important values, such as altruism, service in times of crisis, and solidarity with the profession and disposition to serve society [12]. This opportunity will likely influence the development of professional values and identity.

## Conclusions

Our study showed that medical students who believe that they must participate in the fight against the COVID-19 pandemic are motivated by a sense of purpose or duty, altruism, perception of good performance and values of professionalism more than an fgt in learning. These results have implications for the development of volunteering programs and the design of health force policies in the present pandemic and in future health emergencies.

## Supporting information

**S1 Table. Distribution of answers by each statement for all participants (n = 10,433).**
(DOCX)

**S2 Table. Student characteristics and perceptions according to their views about the role of medical students during the COVID-19 pandemic.** Students were divided into three groups according to their opinions about the participation of medical students in the care of patients with COVID-19. The number and percentage of medical students who answered "completely agree" or "agree" to each of the statements is presented.
(DOCX)

**S3 Table. Relevant items in exploratory factor analysis scores calculation.**
(DOCX)

**S4 Table. Crude odds ratios (95% confidence intervals) for the association between students' characteristics and perceptions and their views on the role of medical students during the COVID-19 pandemic.**
(DOCX)

**S5 Table. Crude odds ratios (95% confidence intervals) for the association between internship students' characteristics and perceptions and their views on the role of medical students during the COVID-19 pandemic.**
(DOCX)

**S1 Questionnaire. Medical students during the COVID-19 pandemic.**
(PDF)

**S2 Questionnaire. O estudante de medicina na pandemia de Covid-19.**
(PDF)

**S1 Dataset.**
(XLSX)

## Acknowledgments

The authors wish to thank the Brazilian Section of the International Federation of Medical Students Associations (IFMSA) for the help during data collection of the study.

## Author Contributions

**Conceptualization:** Patricia Tempski, Fernanda M. Arantes-Costa, Marina A. M. Siqueira, Matheus B. Torsani, Bianca Q. R. C. Amaro, Maria Eduarda F. M. Nascimento, Saulo L. Siqueira, Milton A. Martins.

**Data curation:** Itamar S. Santos.

**Formal analysis:** Patricia Tempski, Renata Kobayasi, Itamar S. Santos, Milton A. Martins.

**Investigation:** Patricia Tempski, Fernanda M. Arantes-Costa, Marina A. M. Siqueira, Matheus B. Torsani, Bianca Q. R. C. Amaro, Maria Eduarda F. M. Nascimento, Saulo L. Siqueira, Milton A. Martins.

**Methodology:** Marina A. M. Siqueira, Matheus B. Torsani, Bianca Q. R. C. Amaro, Maria Eduarda F. M. Nascimento, Itamar S. Santos, Milton A. Martins.

**Software:** Fernanda M. Arantes-Costa.

**Supervision:** Milton A. Martins.

**Validation:** Milton A. Martins.

**Writing – original draft:** Patricia Tempski, Renata Kobayasi, Itamar S. Santos, Milton A. Martins.

**Writing – review & editing:** Patricia Tempski, Fernanda M. Arantes-Costa, Renata Kobayasi, Marina A. M. Siqueira, Matheus B. Torsani, Bianca Q. R. C. Amaro, Maria Eduarda F. M. Nascimento, Saulo L. Siqueira, Itamar S. Santos, Milton A. Martins.

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
