## [Decision Letter · Decision Letter 0]

15 Sep 2020

PONE-D-20-15134

Medical student's perceptions and motivations in time of COVID-19 pandemic

PLOS ONE

Dear Dr. Martins,

Thank you for submitting your manuscript to PLOS ONE. After careful consideration, we feel that it has merit but does not fully meet PLOS ONE’s publication criteria as it currently stands. Therefore, we invite you to submit a revised version of the manuscript that addresses the points raised during the review process.

We look forward to receiving your revised manuscript.

Kind regards,

Janhavi Ajit Vaingankar

Academic Editor

PLOS ONE

Journal Requirements:

3. Please include additional information regarding the survey or questionnaire used in the study and ensure that you have provided sufficient details that others could replicate the analyses.

For instance, if you developed a questionnaire as part of this study and it is not under a copyright more restrictive than CC-BY, please include a copy, in both the original language and English, as Supporting Information. Moreover, please include more details on how the questionnaire was generated and  pre-tested, and whether it was validated; and on how it was disseminated.

5. Please include captions for your Supporting Information files at the end of your manuscript, and update any in-text citations to match accordingly. Please see our Supporting Information guidelines for more information: http://journals.plos.org/plosone/s/supporting-information

Reviewers' comments:

Reviewer's Responses to Questions

**Comments to the Author**

1. Is the manuscript technically sound, and do the data support the conclusions?

Reviewer #1: Yes

Reviewer #2: Yes

Reviewer #3: Partly

2. Has the statistical analysis been performed appropriately and rigorously? 

Reviewer #1: Yes

Reviewer #2: Yes

Reviewer #3: Yes

3. Have the authors made all data underlying the findings in their manuscript fully available?

Reviewer #1: Yes

Reviewer #2: Yes

Reviewer #3: No

4. Is the manuscript presented in an intelligible fashion and written in standard English?

Reviewer #1: Yes

Reviewer #2: No

Reviewer #3: Yes

5. Review Comments to the Author

Reviewer #1: Important study, examining medical students' motives to participate in COVID19 emergency services. Valuable in looking at students as people with their own values, not just at available labor, and motivated by altruism, not just a wish to advance clinical knowledge. Well-designed and clearly presented.

Reviewer #2: I want to start with a positive comment on the high return rate the authors received from the participants. I do think this is a strong research article; however, I would suggest the authors explore some analysis options to verify their assumptions.

My only major piece of feedback is to have the authors conduct a factor analysis of their predictor variables for a couple of reasons: 1. lesson the number of predictors in their logistic regression, and 2. Confirm that the variables measure the same underlying construct they hypothesize.

On line 137, I would change the beginning of the sentence to read: The participants indicated their degree of agreement with ...

This next comment may be a language difference, on page 15, line 167. I am unclear as to what "vacancies" means. Is it the number of non-respondents?

Page 27, line 312. I think using just the term "self-efficacy" is too broad within the context of the variables you used. I would add an adjective to better capture what you think the variables you used to contextualize them specific to the study. For example, clinical self-efficacy, or medical knowledge self-efficacy (a little wordy), etc.

Reviewer #3: The area of research is relevant in the current healthcare landscape. The article is generally well written. However, a number of key components needed to assess scientific merit and infer findings need clarification and elaboration.

1. The aim of the study should be clearly specified. There are a number of tables presented - distribution by year of graduation, gender. The division into 3 groups and analysis thereof was particularly not clear to me. If the aims were clearly specified, it would make things easier for me to understand.

2. The authors mention that the study questionnaire was specifically designed for this study. However I would prefer to get more information on how it was designed. For example, who were the members of the panel and were the decision guided by public health needs, research questions, other similar surveys, etc

Whether any specific domains of interest were tested in these 28 statements. If not, adding some explanation on why as many attitudes would be important to study would be relevant.

3. Response rates seem to be determined by the geography, with better response among southeast regions. Authors could comment on measures they took to ensure uniform response rates and possibly comment in the discussion on how representative their results might be.

4. Analysis is appropriate. However, it is necessary to state why the specific variables were selected for the regression analyses. Much of the rationale for grouping or conducting specific analyses are mentioned later on in the discussion. I feel this needs to be explained before the results to understand the significance of the presented data.

5. I found the tables rather lengthy although I understand that this is due to the length of the developed questionnaire. Perhaps the authors may want to be selective on the amount of information shared in tables within the manuscript and possibly as supplementary tables. Flow of text under results section needs to be improved.

Regarding tables 3-5, I wasnt sure why the items are presented in the random sequence. It may be useful to keep to a ascending sequence based on items number to allow readers to easily compare the results.

6. Discussion is generally well written. Authors however need to state limitations of their research and ways in which these might have influenced the findings.

I would urge the authors to look at standard guidelines, for example STROBE Cross-sectional study checklist that provides a guide on reporting of relevant study components and methodology.

6. PLOS authors have the option to publish the peer review history of their article (what does this mean?). If published, this will include your full peer review and any attached files.

Reviewer #1: No

Reviewer #2: No

Reviewer #3: No

---

## [Author Response · Author response to Decision Letter 0]

30 Oct 2020

Thank you for your letter giving us the opportunity to submit a revised version of our manuscript. We addressed all the points raised during the review process.

We included in Methods more details regarding the questionnaire used to provide sufficient details that others could replicate the analyses. We also included in Methods more details on how the questionnaire was generated and pre-tested.

We also included a file with all data used in the study and there are no legal or ethical restrictions to other researchers use this data set.

We also sent the revised manuscript to American Journal Experts for a language review.

Response to Reviewers

For the convenience of the Editor and the Reviewers, we have retyped the questions and criticisms of the Reviewers.

Reviewer #1: 

Important study, examining medical students' motives to participate in COVID19 emergency services. Valuable in looking at students as people with their own values, not just at available labor, and motivated by altruism, not just a wish to advance clinical knowledge. Well-designed and clearly presented.

Reviewer #2: 

I want to start with a positive comment on the high return rate the authors received from the participants. I do think this is a strong research article; however, I would suggest the authors explore some analysis options to verify their assumptions.

My only major piece of feedback is to have the authors conduct a factor analysis of their predictor variables for a couple of reasons: 1. lesson the number of predictors in their logistic regression, and 2. Confirm that the variables measure the same underlying construct they hypothesize.

Response: Thank you for the suggestion. We included in the manuscript the factor analysis of the questionnaire.

We performed an exploratory factor analysis (EFA) using varimax rotation to identify underlying latent variables in our questionnaire. The criteria for EFA model selection were (1) models with sum of squared loadings above 1 for all latent variables were initially considered and (2) among these models, we selected the model with the highest number of factors. This led to the selection of a model with four factors. Items with loadings below -0.3 or above 0.3 were considered as relevant for each factor. To compute the values for each latent variable, we included the loadings from all items, whether they met or not the criteria for relevance.

Factor analysis resulted in four domains: remote learning, medical knowledge self-efficacy, psychological stress and professional values.

We included in the supplementary material a new table, with the results of the factor analysis (Supplemental Table 2).

We also included the four domains in the regression analysis and included the results in Table 4. 

On line 137, I would change the beginning of the sentence to read: The participants indicated their degree of agreement with ...

Response: we made this change in the manuscript.

This next comment may be a language difference, on page 15, line 167. I am unclear as to what "vacancies" means. Is it the number of non-respondents?

Response: Vacancies in the text mean number of positions for first-year medical students. We changed the text to make this point clear.

Page 27, line 312. I think using just the term "self-efficacy" is too broad within the context of the variables you used. I would add an adjective to better capture what you think the variables you used to contextualize them specific to the study. For example, clinical self-efficacy, or medical knowledge self-efficacy (a little wordy), etc.

Response: We agree and modified the text to “medical knowledge self-efficacy”.

Reviewer #3: 

The area of research is relevant in the current healthcare landscape. The article is generally well written. However, a number of key components needed to assess scientific merit and infer findings need clarification and elaboration.

1. The aim of the study should be clearly specified. There are a number of tables presented - distribution by year of graduation, gender. The division into 3 groups and analysis thereof was particularly not clear to me. If the aims were clearly specified, it would make things easier for me to understand.

Response: The aim of our study was to evaluate motivation of medical students to be part of the heath team to help in the COVID-19 pandemic. To evaluate the motivation of medical students, we decided to develop a questionnaire specifically designed for this purpose.

To perform a regression analysis, we divided the medical students in three groups, according to their opinion regarding who, in their opinion, should participate in the care of patients with COVID-19. The three groups were divided considering the following:

a) Students that responded that medical students should not participate;

b) Students that responded that only students in their final years of medical school should participate (only interns);

c) Students the responded that all medical students should participate in the care of people with COVID-19 (all medical students).

2. The authors mention that the study questionnaire was specifically designed for this study. However, I would prefer to get more information on how it was designed. For example, who were the members of the panel and were the decision guided by public health needs, research questions, other similar surveys, etc. Whether any specific domains of interest were tested in these 28 statements. If not, adding some explanation on why as many attitudes would be important to study would be relevant.

Response: We agree and included in Methods a more detailed description of how the questionnaire was designed. After a careful search in the literature, with did not find any questionnaire designed to assess motivations of medical students to help in the care of people during a pandemic or any other health emergency. So, we decided to develop a new questionnaire.

We performed four meetings, two with medical students and two with faculty from the Center for Development of Medical Education of our medical school. Each meeting had 10-15 participants and lasted 2-3 hours. We asked the following questions to the participants:

“To assess motivations of medical students to work with the health team in the care of people with COVID-19, which questions should be asked?”

“What are the main concerns of medical students related to work with people with COVID-19?”

“What questions should be asked to evaluate the opinions and motivations of medical students concerning the need to move to on-line teaching due to the impact of COVID-19?”

We recorded the meetings, and three researchers wrote the questionnaire including all suggestions from the medical students and faculty meetings.

The questionnaire was then revised by a group of medical students and faculty, until a consensus was reached that all the statements were easy to understand.

A second revision was performed after a pilot application of the questionnaire to a group of twenty medical students.

We included this explanation in Methods

3. Response rates seem to be determined by the geography, with better response among southeast regions. Authors could comment on measures they took to ensure uniform response rates and possibly comment in the discussion on how representative their results might be.

Response: In Table 1 we show the number of respondents from each one of the five geographic regions of Brazil and show the proportion of respondents and number of positions for first-year medical students (vacancies), showing that there was a similar percentage of respondents from each one of the regions. The southeast region of Brazil is the region with more population and more medical schools. We agree that this was not clear in the text and we explained that “vacancies” were the number of first year positions offered to medical students.

4. Analysis is appropriate. However, it is necessary to state why the specific variables were selected for the regression analyses. Much of the rationale for grouping or conducting specific analyses are mentioned later on in the discussion. I feel this needs to be explained before the results to understand the significance of the presented data.

Comments: We included in regression models all 28 statements of the questionnaire as dependent variables (percentages of students that agreed with each one of the 28 statements). We controlled the analysis for sex, year of medical course and region of the country. We included this information in Methods.

We included the rationale for grouping and the analyses in the Methods section.

5. I found the tables rather lengthy although I understand that this is due to the length of the developed questionnaire. Perhaps the authors may want to be selective on the amount of information shared in tables within the manuscript and possibly as supplementary tables. Flow of text under results section needs to be improved.

Comments: we revised the flow of text under results section.

Regarding tables 3-5, I wasn’t sure why the items are presented in the random sequence. It may be useful to keep to an ascending sequence based on items number to allow readers to easily compare the results.

Comments: The items in tables 3-5 are not presented in random sequence. In table 3 “Students characteristics and perceptions according to their view about the role of medical students during the COVID-19 pandemic”, the items are presented in descending sequence concerning the percentage of agreement of each one of the statements of the entire sample of medical students. In tables 4 and 5 (“Adjusted odds ratios for the association between students’ characteristics and perceptions and their view about the role of medical students during the COVID-19 pandemic” and “Adjusted odds ratios for the association between internship students’ characteristics and perceptions and their view about the role of medical students during the COVID-19 pandemic”), the items are presented in descending sequence concerning the odds ratios comparing the group “all medical students should participate in the pandemic” to “no medical students should participate in the pandemic”. We included this information in the Results section of the manuscript. 

6. Discussion is generally well written. Authors however need to state limitations of their research and ways in which these might have influenced the findings.

I would urge the authors to look at standard guidelines, for example STROBE.

Response: We included a paragraph in Discussion about the limitations of our study. We designed our study and report according to STROBE guidelines for studies and included this information in the Methods section of the manuscript.

---

## [Decision Letter · Decision Letter 1]

11 Dec 2020

PONE-D-20-15134R1

Medical student's perceptions and motivations during the COVID-19 pandemic

PLOS ONE

Dear Dr. Martins,

Thank you for submitting your manuscript to PLOS ONE. After careful consideration, we feel that it has merit but does not fully meet PLOS ONE’s publication criteria as it currently stands. Therefore, we invite you to submit a revised version of the manuscript that addresses the points raised during the review process.

We look forward to receiving your revised manuscript.

Kind regards,

Janhavi Ajit Vaingankar

Academic Editor

PLOS ONE

Reviewers' comments:

Reviewer's Responses to Questions

**Comments to the Author**

1. If the authors have adequately addressed your comments raised in a previous round of review and you feel that this manuscript is now acceptable for publication, you may indicate that here to bypass the “Comments to the Author” section, enter your conflict of interest statement in the “Confidential to Editor” section, and submit your "Accept" recommendation.

Reviewer #2: All comments have been addressed

Reviewer #3: 

2. Is the manuscript technically sound, and do the data support the conclusions?

Reviewer #2: Yes

Reviewer #3: Partly

3. Has the statistical analysis been performed appropriately and rigorously? 

Reviewer #2: Yes

Reviewer #3: Yes

4. Have the authors made all data underlying the findings in their manuscript fully available?

Reviewer #2: Yes

Reviewer #3: Yes

5. Is the manuscript presented in an intelligible fashion and written in standard English?

Reviewer #2: Yes

Reviewer #3: Yes

6. Review Comments to the Author

Reviewer #2: The authors addressed all my comments. They correctly pointed out that I made a mistake in my comments about factor analysis.

Reviewer #3: Authors seem to have addressed most of the suggestions. I would however like to provide few further comments for consideration:

1. Methods section could be made clearer by providing a proper flow to the content. Perhaps adding sub sections such as ethics, study setting, participant criteria, development of questionnaire and data collection could offer clarity to the readers.

2. Statistical section: The factor structure was performed after suggestions in earlier reviewer. However, it is left at that. Authors should consider how the 4 factors could be used to score the responses and then use them to perform regression analysis for the subscales. The tables are currently very lengthy and far too much data is presented. The item-wise distribution does not add much value as there aren't many evident differences in the three groups. I would urge the authors to just focus on subscale scores for some of the analysis and their determinants instead of item score differences.

3. The discussion needs to clearly address the aim of the study which was to identify motivations of the students to be part of pandemic teams. A summary of the main motivations or unexpected findings could be provided at the beginning. This can also help in understanding the multitude of results presented.

7. PLOS authors have the option to publish the peer review history of their article (what does this mean?). If published, this will include your full peer review and any attached files.

Reviewer #2: No

Reviewer #3: No

---

## [Author Response · Author response to Decision Letter 1]

6 Jan 2021

Manuscript PONE-D-20-15134R1

“Medical students' perceptions and motivations during the COVID-19 pandemic”

Point-by-point reply to the comments of Reviewer #3

Dear Dr. Janhavi Ajit Vaingankar

Academic Editor

PLOS ONE

Thank you for your letter with the invitation to submit a revised version of our manuscript addressing the points raised by reviewer #3.

For your convenience we have retyped the comments of the reviewers.

Reviewer #2: The authors addressed all my comments. They correctly pointed out that I made a mistake in my comments about factor analysis.

Reviewer #3: Authors seem to have addressed most of the suggestions. I would however like to provide few further comments for consideration:

1. Methods section could be made clearer by providing a proper flow to the content. Perhaps adding sub sections such as ethics, study setting, participant criteria, development of questionnaire and data collection could offer clarity to the readers.

Response: We added sub sections to Methods, as suggested: ethics, development of questionnaire, data collection, study variables and statistical analysis.

2. Statistical section: The factor structure was performed after suggestions in earlier reviewer. However, it is left at that. Authors should consider how the 4 factors could be used to score the responses and then use them to perform regression analysis for the subscales. The tables are currently very lengthy and far too much data is presented. The item-wise distribution does not add much value as there aren't many evident differences in the three groups. I would urge the authors to just focus on subscale scores for some of the analysis and their determinants instead of item score differences.

Response: We included the factor analysis in the revised version of the manuscript and included the four domains identified in the multinomial logistic regression models to study the association between the students’ characteristics and perceptions according to their opinion about student participation in the healthcare of the COVID-19 pandemic (Table 3: Adjusted odds ratios for the association between students’ characteristics and perceptions and their views on the role of medical students during the COVID-19 pandemic). We included both the four domains and all statements of the questionnaire in the logistic regressions.

To decrease the amount of data presented in the main manuscript we moved former table 3 (“Student characteristics and perceptions according to their views about the role of medical students during the COVID-19 pandemic”) to supplementary material.

3. The discussion needs to clearly address the aim of the study which was to identify motivations of the students to be part of pandemic teams. A summary of the main motivations or unexpected findings could be provided at the beginning. This can also help in understanding the multitude of results presented.

Response: We agree and included a new paragraph in the beginning of Discussion (paragraph #2) with the summary of the main motivations of medical students observed).

---

## [Decision Letter · Decision Letter 2]

9 Feb 2021

PONE-D-20-15134R2

Medical students' perceptions and motivations during the COVID-19 pandemic

PLOS ONE

Dear Dr. Martins,

Thank you for submitting your manuscript to PLOS ONE. After careful consideration, we feel that it has merit but does not fully meet PLOS ONE’s publication criteria as it currently stands. Therefore, we invite you to submit a revised version of the manuscript that addresses the points raised during the review process.

We look forward to receiving your revised manuscript.

Kind regards,

Janhavi Ajit Vaingankar

Academic Editor

PLOS ONE

Reviewers' comments:

Reviewer's Responses to Questions

**Comments to the Author**

1. If the authors have adequately addressed your comments raised in a previous round of review and you feel that this manuscript is now acceptable for publication, you may indicate that here to bypass the “Comments to the Author” section, enter your conflict of interest statement in the “Confidential to Editor” section, and submit your "Accept" recommendation.

Reviewer #3: All comments have been addressed

2. Is the manuscript technically sound, and do the data support the conclusions?

Reviewer #3: Partly

3. Has the statistical analysis been performed appropriately and rigorously? 

Reviewer #3: Yes

4. Have the authors made all data underlying the findings in their manuscript fully available?

Reviewer #3: Yes

5. Is the manuscript presented in an intelligible fashion and written in standard English?

Reviewer #3: Yes

6. Review Comments to the Author

Reviewer #3: Authors have addressed my comments well. However, I have one further comment regarding the scores derived from the four factors. Authors should add how the scores were obtained (eg total scores, weighted scores etc) in the analysis section. This will make it easier to interpret the ORs presented in Table 3. I can understand the categorical responses for each item and the reference used therein. But I am not clear how from each factor (eg. altruism) a categorical response was obtained? I gather there were more than 1 item in each factor? Some clarity on this will be useful.

7. PLOS authors have the option to publish the peer review history of their article (what does this mean?). If published, this will include your full peer review and any attached files.

Reviewer #3: No

---

## [Author Response · Author response to Decision Letter 2]

15 Feb 2021

Manuscript PONE-D-20-15134R2

“Medical students’ perceptions and motivations during the COVID-19 pandemic”

Point-by-point reply to the comments of Reviewer #3

Dear Dr. Janhavi Ajit Vaingankar

Academic Editor

PLOS ONE

Thank you for your letter with the invitation to submit a revised version of our manuscript addressing the points raised by reviewer #3.

Reviewer #3: Authors have addressed my comments well. However, I have one further comment regarding the scores derived from the four factors. Authors should add how the scores were obtained (eg total scores, weighted scores etc) in the analysis section. This will make it easier to interpret the ORs presented in Table 3. I can understand the categorical responses for each item and the reference used therein. But I am not clear how from each factor (eg. altruism) a categorical response was obtained? I gather there were more than 1 item in each factor? Some clarity on this will be useful.

Answer to reviewer #3: Thank you for your suggestion. In the new version of the paper, we clarified the interpretation of odds ratios for exploratory factor analysis (EFA) latent variables. EFA latent variables are included in regression models as continuous independent variables. In the new version, to improve interpretation and comparability, we standardized the scores for each EFA latent variable, adopting a mean of zero and standard deviation of one. Therefore, the odds ratios presented in the paper relate to a 1 SD increase in the factor score.

We included in Methods:

For exploratory factor analysis (EFA) and regression models, agreement with each statement was considered as a dichotomous variable. In these cases, participants who answered “totally agree” or “agree” were considered as agreeing with each statement. We performed an EFA using varimax rotation to identify the underlying latent variables (factors) in our questionnaire. The criteria for EFA model selection were as follows: (1) models with sum of squared loadings greater than 1 for all latent variables were initially considered, and (2) among these models, we selected the model with the highest number of factors. This process led to the selection of a model with four factors. Items with loadings less than -0.3 or greater than 0.3 were considered relevant for each factor (Supplemental table 1). To compute each EFA latent variable score, we attributed weights for all items, according to their loadings in the final model, regardless of whether they met the criteria for relevance. After this procedure, we standardized the values to obtain a mean of zero and standard deviation (SD) of one for each factor score. Therefore, the odds ratios presented in these models relate to a 1 SD increase in the EFA latent variable score.

---

## [Editor Report · Decision Letter 3]

17 Feb 2021

PONE-D-20-15134R3

Medical students' perceptions and motivations during the COVID-19 pandemic

PLOS ONE

Dear Dr. Martins,

Thank you for submitting your manuscript to PLOS ONE. After careful consideration, we feel that it has merit but does not fully meet PLOS ONE’s publication criteria as it currently stands. Therefore, we invite you to submit a revised version of the manuscript that addresses the points raised during the review process.

In revising your submission, please carefully address these additional comments:

1. Please elaborate/clarify how the analyses on gender differences were conducted. Table 3 and 4 present multiple reference categories and it is unclear whether multiple regression models were tested. You may also want to edit the following sentence in the abstract accordingly: "Males exhibited higher odds ratios than females (1.36 [95% CI: 1.24 - 1.49] versus 1.68 [95% CI: 1.47 – 1.91])". 

2. Please also check all rows and columns in the tables to ensure correct reference groups are mentioned.

3. It will be useful for the readers if authors can add interpretations of the ORs resulting from EFA-based scores. eg what does 1 SD difference in OR mean in terms of the students' attitudes?

We look forward to receiving your revised manuscript.

Kind regards,

Janhavi Ajit Vaingankar

Academic Editor

PLOS ONE

---

## [Author Response · Author response to Decision Letter 3]

21 Feb 2021

PONE-D-20-15134R3

“Medical students' perceptions and motivations during the COVID-19 pandemic”

Point-by-point reply

For the convenience of the Academic Editor, we have retyped all comments of the Academic Editor.

In revising your submission, please carefully address these additional comments:

1. Please elaborate/clarify how the analyses on gender differences were conducted. Table 3 and 4 present multiple reference categories and it is unclear whether multiple regression models were tested. You may also want to edit the following sentence in the abstract accordingly: "Males exhibited higher odds ratios than females (1.36 [95% CI: 1.24 - 1.49] versus 1.68 [95% CI: 1.47 – 1.91])".

Reply: We modified Discussion to better explain how gender differences were evaluated. We performed regression analysis comparing the number of male and female students that agreed with the three statements concerning participation of medical students in the care of people with COVID-19 pandemic: “no students should participate”, “only interns should participate” and “all medical students should participate”.

We modified Abstract to better explain gender differences observed:

“Males were more prone than females to believe that only interns should participate in the care of patients with COVID-19 (odds ratio 1.36 [coefficient interval 95%:1.24-1.49]) and that all students should participate (OR 1.68 [CI:1.4-1.91]).”

2. Please also check all rows and columns in the tables to ensure correct reference groups are mentioned.

Reply: We checked all rows and columns in all tables of the main manuscript, and also of supplemental material.

3. It will be useful for the readers if authors can add interpretations of the ORs resulting from EFA-based scores. eg what does 1 SD difference in OR mean in terms of the students' attitudes?

Reply: In Methods we better described how exploratory factor analysis was performed and how the results should be interpreted:

“After this procedure, we standardized the values to obtain a mean of zero and standard deviation (SD) of one for each factor score. Therefore, the odds ratios presented in these models relate to a 1 SD increase in the EFA latent variable score. In this model, an odds ratio of 2 represents twice the chance of a respondent have a score one standard deviation higher than the mean score of the factor.”

---

## [Editor Report · Decision Letter 4]

3 Mar 2021

Medical students' perceptions and motivations during the COVID-19 pandemic

PONE-D-20-15134R4

Dear Dr. Martins,

We’re pleased to inform you that your manuscript has been judged scientifically suitable for publication and will be formally accepted for publication once it meets all outstanding technical requirements.

Kind regards,

Janhavi Ajit Vaingankar

Academic Editor

PLOS ONE

---

## [Editor Report · Acceptance letter]

8 Mar 2021

PONE-D-20-15134R4 

Medical students’ perceptions and motivations during the COVID-19 pandemic 

Dear Dr. Martins:

I'm pleased to inform you that your manuscript has been deemed suitable for publication in PLOS ONE. Congratulations! Your manuscript is now with our production department. 

Kind regards, 

on behalf of

Ms Janhavi Ajit Vaingankar 

Academic Editor

PLOS ONE